# Supramolecular Functionalisation of B/N Co-Doped Carbon Nano-Onions for Novel Nanocarrier Systems

**DOI:** 10.3390/ma15175987

**Published:** 2022-08-30

**Authors:** Hugh Mohan, Valeria Bincoletto, Silvia Arpicco, Silvia Giordani

**Affiliations:** 1School of Chemical Sciences, Dublin City University, Glasnevin, D09 NA55 Dublin, Ireland; 2Department of Drug Science and Technology, University of Torino, Via P. Giuria 9, 10125 Torino, Italy

**Keywords:** carbon nano-onions, boron/nitrogen doping, hyaluronic acid, nanocarrier, phospholipid, supramolecular system, CD44, carbon nanomaterials, aqueous dispersibility, conjugate polymer

## Abstract

Boron/nitrogen co-doped carbon nano-onions (BN-CNOs) are spherical nanoparticles that consist of multiple inter-nestled fullerene layers, giving them an onion-like internal structure. They have potential as nanocarriers due to their small size, aqueous dispersibility, and biocompatibility. The non-covalent attachment of a biocompatible polymer to BN-CNOs is a simple and effective method of creating a scaffold for a novel nanocarrier system as it allows for increased aqueous dispersibility whilst preventing the immune system from recognising the particle as a foreign object. The non-covalent approach also preserves the electronic and structural properties of the BN-CNOs. In this study, we attached a hyaluronic acid-phospholipid (HA-DMPE) conjugate polymer to the BN-CNO’s surface to improve its hydrophilicity and provide targetability toward HA-receptor overexpressing cancer cells. To this end, various ratios of HA-DMPE to BN-CNOs were investigated. The resulting supramolecular systems were characterised via UV-Vis absorption and FTIR spectroscopy, dynamic light scattering, and zeta potential techniques. It was found that the HA-DMPE conjugate polymer was permanently wrapped around the BN-CNO nanoparticle surface. Moreover, the resulting BN-CNO/HA-DMPE supramolecular systems displayed enhanced aqueous solubility compared to unfunctionalised BN-CNOs, with excellent long-term stability observed in aqueous dispersions.

## 1. Introduction

First discovered by Ugarte in 1992 [1], carbon nano-onions (CNOs) are spherical nanoparticles that consist of multiple fullerene layers nestled together, giving them an onion-like internal structure. CNOs exhibit unique electronic properties, excellent thermal stability, broad absorption spectra, and a large surface area [2]. Depending on the synthesis method used, CNO size can be controlled, with CNOs produced with a diameter from 1.4 to 100 nm [3]. These production protocols include arc-discharge [4], laser vaporisation [5], thermal annealing [6], and thermolysis [7]. The addition of dopant atoms such as nitrogen [8] and boron [9] to the CNO structure adds an additional layer of customisation, enabling the optimisation of CNO structure and composition for particular applications. In particular, the doping of CNOs with heteroatoms can impart luminescence [10], sensing [11], and energy storage [12] properties. 

Our group has developed a new type of CNO; the boron/nitrogen co-doped carbon nano-onion (BN-CNO). This nanomaterial shows enhanced aqueous dispersibility and low cytotoxicity in both in vitro and in vivo studies [13]. These BN-CNOs are produced through a simple, low-cost, and environmentally friendly method involving the thermal annealing of detonation nano-diamonds in the presence of boric acid [14]. The BN-CNOs have potential as nanocarriers due to their small size and aforementioned biocompatibility, whilst also displaying increased aqueous dispersibility over regular CNOs.

Although CNOs are biocompatible [15], a significant drawback to their deployment in biological systems is their hydrophobicity, similar to other carbon nanomaterials [16]. Although CNOs can be dispersed in organic solvents, they can still form aggregates in solution [17]. For this reason, the surface of CNOs is commonly functionalised to increase hydrophilicity. A wide range of surface chemistry has also been developed for CNOs [18,19]. However, covalent modification methods, such as oxidation, are known to damage the structure of CNOs by disturbing the carbon sp^2^ lattice, converting C atoms to sp^3^ hybridisation [20], leading to a loss of their unique electronic and physical properties. Although BN-CNOs have improved aqueous dispersibility compared to CNOs due to the increased presence of oxygen-containing functional groups combined with heteroatoms in the pristine form (Figure 1 Framed), they too require further functionalisation for true long-term aqueous dispersibility. In addition, the oxidation of BN-CNOs results in a loss of heteroatom content due to the preferential oxidation of the active B/N sites [13]. Therefore, this study focuses on the non-covalent surface functionalisation of CNOs.

The coating used in this study is a 200 kDa hyaluronic acid-1,2-dimyristoyl-*sn*-glycero-3-phosphoethanolamine (HA-DMPE) conjugate polymer (Figure 1 Top). HA and DMPE are biocompatible molecules. The purpose of using HA-DMPE is twofold; the conjugate is used to improve the aqueous dispersibility of the BN-CNOs by acting as a surfactant. The aliphatic chains of the DMPE phospholipid non-covalently bind to the BN-CNO surface via hydrophobic interactions; this type of bonding preserves the structure of the nano-onions by avoiding damage to the sp^2^ lattice that generally results from covalent modification. The plentiful hydroxyl, ether, and carboxylic acid groups in HA interact with water to increase the hydrophilicity of the nanocomposite. Besides its dispersant properties, HA is used for its targetability [21,22,23]. HA acts as a targeting ligand towards cancer cells overexpressing the hyaluronate (CD44) receptor [24]. Furthermore, HA can avoid the adsorption of many blood proteins, prevent coagulation, and prolong the half-life of HA-decorated nanoparticles in the circulatory system [25]. In a relevant study, our group produced a CNO/HA-DMPE nanocomposite that exhibited aqueous stability, biocompatibility, cellular uptake and excretion, and preferential targeting of CD44^+^ cells [26].

Polymer surfactant/nanoparticle dispersions can exist as either dynamic or static dispersions, as seen in (Figure 1 Bottom). In dynamic dispersions utilising head/tail surfactants, such as sodium dodecyl sulphate, the dispersant can exist in equilibrium with the surrounding solvent. This reversible binding can cause issues during biological studies, as the equilibrium can shift in the bloodstream/cellular matrix due to dilution, resulting in nanoparticle re-agglomeration. As for static dispersions, a polymer wrapping mechanism is utilised to bind the surfactant to the nanoparticle surface. This mechanism has the advantage of producing a thermodynamically stable polymer coating that is not removed upon dilution [27]. With static dispersions, excess surfactant can be removed via filtration, centrifugation, or dialysis without affecting the functionalised nanoparticles, which circumnavigates the toxicity issues often associated with dispersants [28].

This study aimed to create a BN-CNO/HA-DMPE hybrid nanocarrier system via a non-covalent polymer wrapping mechanism. This nanocarrier system incorporates a novel BN-CNO scaffold, which has improved aqueous dispersibility compared to undoped carbon nanomaterials. This system also incorporates HA-DMPE, a biocompatible polymer conjugate with targetability toward the CD44 receptor. The synthesis of the supramolecular system and its components is simple and environmentally friendly, and the individual components are biocompatible [13,26]. Physicochemical characterisation was performed on BN-CNO/HA-DMPE dispersions using UV-Vis absorption, FTIR, and DLS methods. The long-term stability of aqueous dispersions of the supramolecular system was determined, and the nature of the HA-DMPE binding was investigated.

## 2. Materials and Methods

### 2.1. Materials

Nano-diamond powder (uDiamond^®^ Molto, 4.2 ± 0.5 nm crystal size) was purchased from Carbodeon (Pakkalankuja, Finland). Boric acid (≥99.5%) was purchased from Sigma Aldrich (Arklow, Ireland). The phospholipid 1,2-dimyristoyl-*sn-*glycero-3-phosphoethanolamine (DMPE) was purchased from Merck (Milan, Italy). A quantity of 200 kDa sodium hyaluronate (HA) was purchased from Lifecore Biomedical (Chaska, MN, USA).

### 2.2. Synthesis of BN-CNOs

Boron/nitrogen co-doped carbon nano onions (BN-CNOs) were synthesised according to a previously reported procedure involving the thermal annealing of detonation nano-diamonds in the presence of boric acid [14].

### 2.3. Synthesis of HA-DMPE

The 200 kDa hyaluronic acid-1,2-dimyristoyl-*sn*-glycero-3-phosphoethanolamine (HA-DMPE) conjugate polymer was synthesised as previously described [26].

### 2.4. Non-Covalent Functionalisation of BN-CNOs with HA-DMPE

The surface of BN-CNOs was non-covalently functionalised with the HA-DMPE conjugate. Briefly, BN-CNOs were added to a vial with water to make a concentration of 1 mg/mL. Enough solid HA-DMPE was added to give 2:1, 5:1, and 10:1 BN-CNO/HA-DMPE mass ratios, respectively. The solution was sonicated for 1.5 h using an Elmasonic S30 ultrasonic bath; the water temperature was maintained under 30 °C.

The washed sample was prepared by vacuum filtering 100 mL of a 10:1 100 µg/mL dispersion using a 0.2 µm nylon filter. The material was then washed by passing 300 mL of deionised water through the vacuum filter. The filter papers were then dried in a vacuum oven (800 mbar, 30 °C) for 24 h. After carefully scraping the functionalised material off, it was weighed and re-constituted to its original concentration by sonicating in deionised water for 30 min.

### 2.5. UV-Vis Absorption Spectroscopy

UV-Vis absorption analysis was carried out using a Shimadzu UV-1800 instrument with 1 cm path-length quartz cuvettes. The 1 mg/mL dispersions from the functionalisation procedure were diluted to 100 µg/mL using deionised water and sonicated for 15 min, then further diluted to 50, 20, 10, 5, and 2.5 µg/mL. Absorption spectra were smoothed to reduce the lamp switch noise at ~340 nm.

### 2.6. Dynamic Light Scattering and Zeta Potential

Dynamic light scattering (DLS) and zeta potential analyses were performed using a Malvern Zen 3600 Zetasizer. For DLS, the instrument was set to backscattering mode (173°).

### 2.7. Fourier Transform Infrared Spectroscopy

Measurements of the dried, washed sample were taken using a Thermoscientific Nicolet Summit FTIR instrument with an Everest ATR accessory containing a diamond crystal and a detector with a KBr window. ATR-FTIR spectra were normalised from 0 to 1 for comparative purposes.

## 3. Results

### 3.1. UV-Vis Absorption Spectroscopy

UV-Vis spectroscopy is an excellent tool for determining the dispersing abilities of the 200 kDa hyaluronic acid-1,2-dimyristoyl-*sn*-glycero-3-phosphoethanolamine (HA-DMPE) conjugate polymer. The calibration plots in Appendix A reveal a measurable absorbance at concentrations as low as 2.5 µg/mL for all three mass ratios, indicating that the polymer is an excellent dispersant for boron/nitrogen co-doped carbon nano-onions (BN-CNOs). Interestingly, the molar extinction coefficient (ε) for the 2:1 BN-CNO/HA-DMPE mass ratio was significantly higher than those of the 5:1 and 10:1 samples, which were close in magnitude (Appendix A).

Although the initial dispersion of the BN-CNOs with HA-DMPE was readily achieved, it is essential to consider the time-dependent stability of these aqueous dispersions. UV-Vis absorption studies (Figure 2) reveal a long-term BN-CNO/HA-DMPE stability at 100 and 50 µg/mL starting concentrations. These stability studies show that the BN-CNO/HA-DMPE system is stable in water and remains dispersed, even after several months. Dispersion stability is an important property that may enable the systems’ future drug delivery applications. The numerical values in Table 1 were obtained by exploiting the Beer–Lambert law; they confirm that, even after 6 months, approximately half of the supramolecular system remains in solution. Concentration values remain similar between mass ratios throughout the study. The 10:1 dispersion exhibited similar stability to the 5:1 dispersion at 50 µg/mL and better stability at the 100 µg/mL concentration. Photos of the 50 µg/mL dispersions can be seen in Figure 3.

### 3.2. Dynamic Light Scattering and Zeta Potential

Regarding the size of the particles in these dispersions, dynamic light scattering (DLS) was used to determine the mean hydrodynamic diameter of each sample. This value correlates to the size of the particles in solution, especially in the case of spherical nanoparticles such as the BN-CNO/HA-DMPE nanoconjugate. This analysis is simple and cheap to perform, making it an ideal characterisation technique for BN-CNO dispersions. Figure 4 shows the effects of varying dispersion concentration (Figure 4a–c) and stand time (Figure 4d–f) on the hydrodynamic diameters of the 2:1, 5:1, and 10:1 BN-CNO/HA-DMPE nanoconjugate.

The DLS spectra in Figure 4a–c show that decreasing a dispersion’s concentration reduces the hydrodynamic diameter. These size values are tabulated in (Table 2), including zeta potentials of each composite measured at 100 µg/mL. The zeta potential average for the three ratios is approx. −30 mV, indicating that the nanoparticles are stable in solution and that re-agglomeration is unfavourable due to electrostatic repulsion between the particles.

Large aggregates (approx. 6 μm in diameter) can be seen in all DLS spectra in Figure 4 (the small peaks to the right of the central peak). These agglomerates likely consist of either BN-CNOs clumps or masses of tangled HA-DMPE polymer chains. The DLS stability spectra in Figure 4d–f show that these aggregates disappear when the dispersions are allowed to stand for a week. Table 3 shows the mean hydrodynamic diameters; the peaks at 270, 286, and 268 nm for the 2:1, 5:1, and 10:1 ratios demonstrate that the sample is well-dispersed. A general downward trend in hydrodynamic radius with time is observed in Table 3, with the 2:1 and 5:1 samples beginning to re-agglomerate at the three-week mark.

### 3.3. Infrared Spectroscopy

Fourier transform infrared spectroscopy (FTIR) is a simple and cost-effective technique that can detect the functional groups on the surface of BN-CNOs. Moreover, FTIR spectroscopy makes it possible to determine whether the HA-DMPE is permanently attached to the BN-CNO surface by taking measurements of the solid and washed BN-CNO/HA-DMPE sample.

Figure 5 depicts the FTIR spectra of HA-DMPE, a washed 10:1 BN-CNO/HA-DMPE sample, and a sample of BN-CNOs. The washed 10:1 BN-CNO/HA-DMPE sample exhibits the following vibrational bands that arise from the HA-DMPE functional groups: OH stretching (3300 cm^−1^), OH bending (1430 cm^−1^), C=O stretching (1605 cm^−1^), and C-O stretching (1093 cm^−1^). These bands arise from the multiple hydroxyl, carbonyl, and ether groups present in the HA-DMPE conjugate polymer, and their presence confirms successful BN-CNO functionalisation. As the sample was washed with a large volume of water, the presence of characteristic HA-DMPE bands suggests that the polymer is permanently functionalised onto the surface of the BN-CNOs.

### 3.4. UV-Vis Absorption and DLS of BN-CNO/HA-DMPE Washed Sample

The UV-Vis stability and DLS analysis presented in Figure 6a,b further confirm that the HA-DMPE is statically attached to the BN-CNO surface. When re-dispersed, the washed sample retains its stability in water and gives a high absorption. The DLS results are similar to that of the original, unwashed sample. The large aggregates seen previously are also still present in the DLS spectra of the washed sample.

## 4. Discussion

UV-Vis absorption stability studies indicate successful B/N co-doped carbon nano-onion (BN-CNO) functionalisation. The non-covalent attachment of 200 kDa hyaluronic acid-1,2-dimyristoyl-*sn*-glycero-3-phosphoethanolamine (HA-DMPE) onto the BN-CNO surface imparted excellent aqueous dispersibility; this result was also previously observed by our group with pristine CNOs [26]. Each sample shows strong absorbance across the entire UV-Vis range, with a prominent peak at around 270 nm (Figure 2). This band arises from π–π* transitions in the C=C bonds of BN-CNOs. In contrast, hyaluronic acid (HA) does not exhibit this peak in its UV-Vis absorption spectrum (Appendix A). Therefore, UV-Vis absorption spectroscopy can be used to confirm the BN-CNO dispersion enabled by the introduction of HA-DMPE. These BN-CNO/HA-DMPE dispersions showed excellent long-term stability, with approximately 50% of the initial nanocomposite remaining in solution after 6 months (Table 1). Interestingly, the 10:1 mass ratio sample performed similarly to, if not better than, the 5:1 and 2:1 dispersions. The ability to disperse the nano-onions with a small amount of dispersant is important, as this may potentially leave space on the surface of the BN-CNOs for future drug and fluorophore attachment. Our group has not previously investigated this low BN-CNO/HA-DMPE ratio.

It is of interest to consider whether it is possible to control the size of this system. It is known that smaller nanoparticles are more likely to enter cells via passive uptake; however, if they are too small, nanoparticles can cause toxicity [29]. It is important to note that, in solution, the BN-CNO/HA-DMPE exist as aggregates and not individual nanoparticles. By performing dynamic light scattering (DLS) analysis, BN-CNO/HA-DMPE supramolecular systems’ hydrodynamic diameter could be assessed (Table 2). It was found that the aggregate size of these particles can be reduced by lowering their concentration. Moreover, by simply letting the dispersions stand undisturbed, the size of the aggregates could also be reduced. Although the nanocomposite particles are small enough to enter cells, the presence of large aggregates is concerning as they may interfere with future in vitro experiments. As seen in Figure 4, these clusters can be removed by letting the dispersions stand for a week. It is also possible that centrifugation could be used to remove these clusters. 

The zeta potential (ZP) values obtained (Table 2)—which averaged approximately −30 mV—indicate that the BN-CNO/HA-DMPE dispersions are stabilised by electrostatic repulsion between nanoparticles. Interestingly, the ZP value obtained for the 5:1 dispersion was 5 mV higher than that obtained for a 5:1 pristine CNO/HA-DMPE dispersion prepared and measured under identical conditions in our previous study [26]. These lower values may be due to a different orientation of the polymer on the BN-CNO surface compared to the pristine CNO surface.

The FTIR results (Figure 5) show the presence of characteristic HA-DMPE vibration bands in the BN-CNO/HA-DMPE supramolecular system, indicating successful non-covalent functionalisation of the BN-CNO surface with the conjugate polymer. Furthermore, FTIR analyses indicate that the BN-CNO/HA-DMPE exist as a static dispersion, meaning the HA-DMPE conjugate polymer is permanently affixed to the nanoparticle’s surface. This observation is further confirmed by an additional stability study performed on the washed, re-dispersed sample, as seen in Figure 6. This 10:1 BN-CNO/HA-DMPE re-dispersed sample demonstrated excellent water dispersibility and stability similar to the original, unwashed sample.

Although the BN-CNO synthesis method is environmentally friendly, it does require the use of artificially created nano-diamonds and the addition of synthetically sourced dopants. A more straightforward “self-doping” approach, such as the method utilised by Yunhao et al. [10], may be incorporated to simplify doped CNO production. This method also has the advantage of using biomass instead of artificially produced diamonds. The nano-onions produced by this method are also fluorescent, without the addition of dyes. A similar fluorescent doped nano-onion material was also produced by Ghosh et al. [30]. Developing fluorescent nano-onions for nanocarrier applications would be advantageous as it would remove the need for dye molecules such as fluoresceinamine for tracking and imaging experiments. This doping approach may simplify the procedure for making such nanocarriers and circumnavigate problems such as fluorescence quenching.

## 5. Conclusions

In this study, boron/nitrogen co-doped carbon nano-onions (BN-CNOs) were non-covalently functionalised with a 200 kDa hyaluronic acid-1,2-dimyristoyl-*sn*-glycero-3-phosphoethanolamine (HA-DMPE) conjugate polymer to impart aqueous dispersibility and targeting affinity for the CD44 receptor, which is overexpressed in many cancer types. Successful binding was confirmed by UV-Vis absorbance and FTIR spectroscopy. The resulting nanocomposite showed excellent long-term aqueous dispersibility. The high stability of a nanocarrier in water at concentrations as high as 100 µg/mL is essential for future use in drug delivery and imaging applications. Dynamic light scattering (DLS) analysis confirmed that the nanocomposite was small enough to enter cells via passive uptake. Controlling the hydrodynamic diameter of the nanoparticles was possible by varying concentrations and letting the dispersion stand undisturbed. Furthermore, it was found that the nanocomposite exists as a static dispersion in water, as the HA-DMPE conjugate polymer is permanently affixed to the surface of BN-CNOs, as confirmed by FTIR and UV-Vis absorption spectroscopy. 

Overall, these results outline the potential of the BN-CNO/HA-DMPE supramolecular system presented herein as a drug delivery vehicle/nanocarrier. Specifically, these results indicate that the system may remain soluble in the body for extended periods, allowing CD44-targeted drug delivery and the subsequent excretion of the nanocarrier. To this end, in vitro investigations are warranted. In summary, this study represents the first non-covalent functionalisation of BN-CNOs and establishes their potential for use as nanocarriers. We envision that a drug payload can be incorporated into the BN-CNO/HA-DMPE nanocarrier scaffold described herein to create a nanocarrier system with selective targetability and controlled drug release. 

## Figures and Tables

**Figure 1 materials-15-05987-f001:**
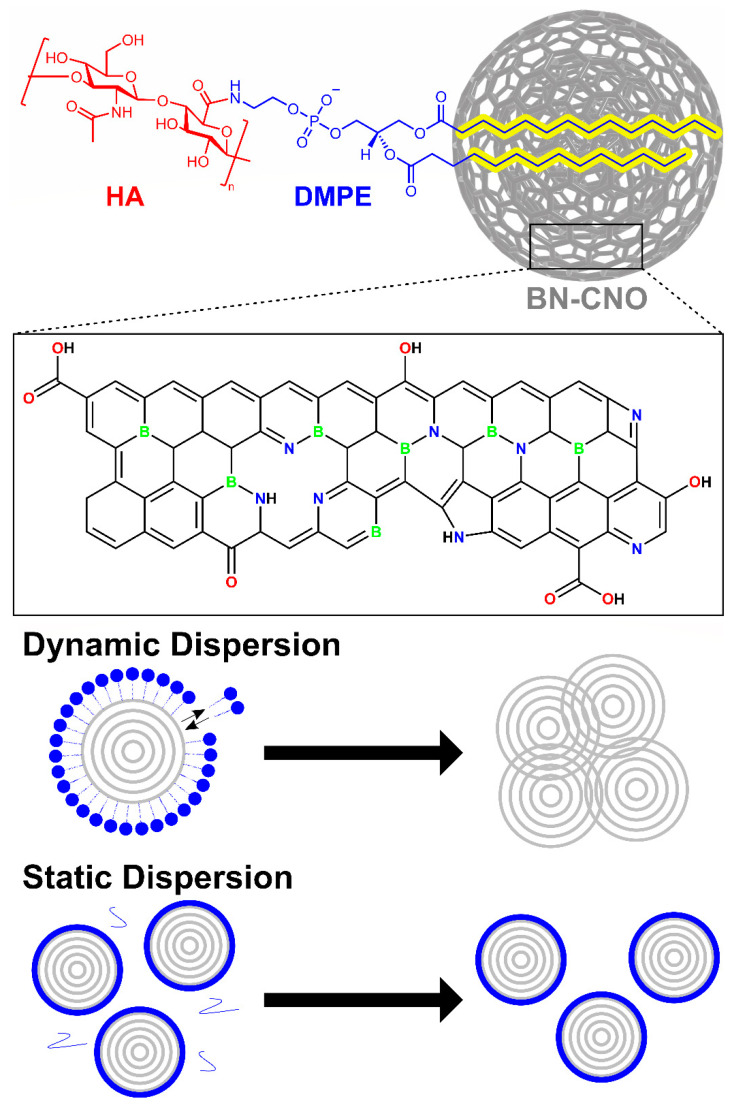
(**Top**) Hyaluronic acid-1,2-dimyristoyl-*sn*-glycero-3-phosphoethanolamine non-covalently bound to the BN-CNO surface. (Framed) The functional groups present on the BN-CNO surface. The size ratio of HA-DMPE to the BN-CNOs presented is for schematic purposes; in reality, the BN-CNOs are much larger than the HA-DMPE polymer chains. They also exist as clusters in solution, not individual particles. (**Bottom**) Illustration of a dynamic dispersion and a static dispersion.

**Figure 2 materials-15-05987-f002:**
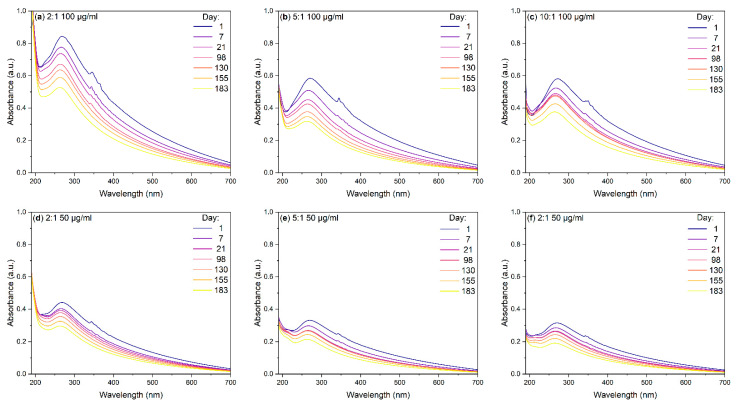
UV-Vis absorption spectra of (**a**) 2:1 BN-CNO/HA-DMPE, (**b**) 5:1 BN-CNO/HA-DMPE, and (**c**) 10:1 BN-CNO/HA-DMPE at various times with a 100 µg/mL starting concentration; (**d**–**f**) = UV-Vis spectra of 2:1, 5:1, and 10:1 BN-CNO/HA-DMPE, respectively, at various times (50 µg/mL starting concentration); solvent = deionised water.

**Figure 3 materials-15-05987-f003:**
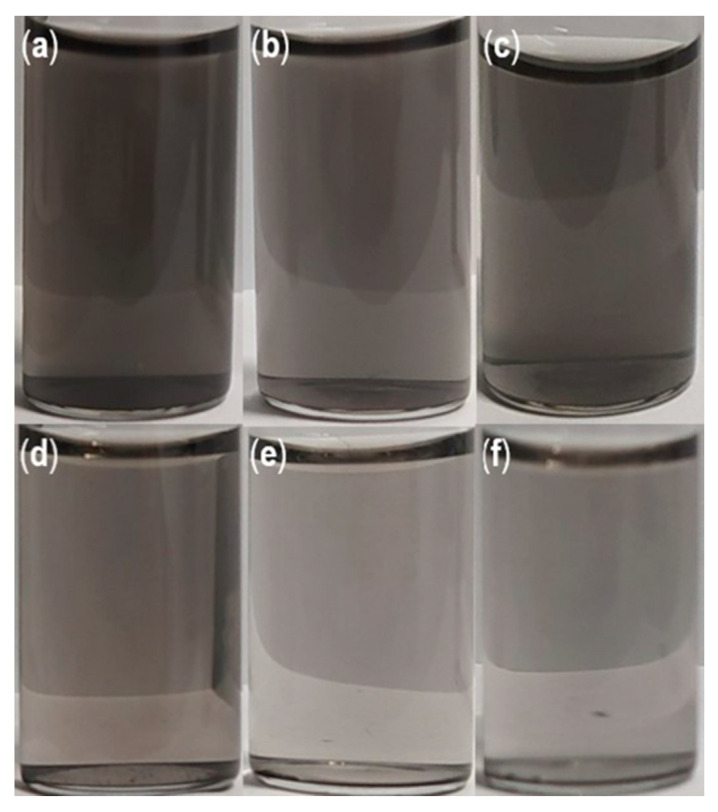
Photos of (**a**–**c**) 2:1, 5:1, and 10:1 BN-CNO/HA-DMPE dispersions straight after sonication; (**d**–**f**) 2:1, 5:1, and 10:1 BN-CNO/HA-DMPE dispersions after 21 days; (solvent = deionised water; 50 µg/mL starting concentration).

**Figure 4 materials-15-05987-f004:**
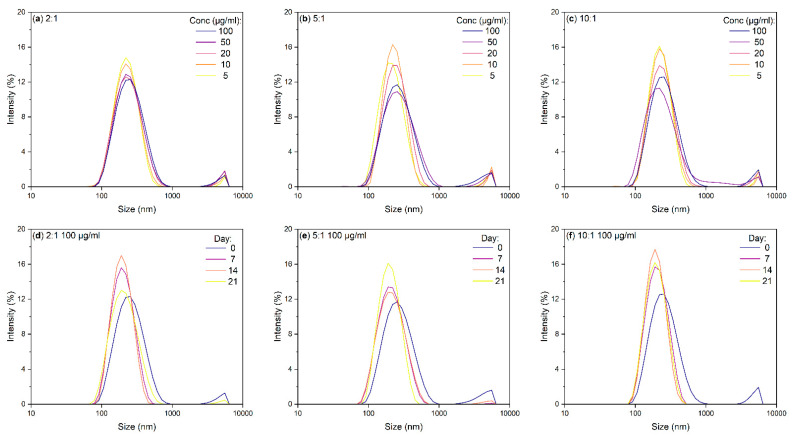
DLS spectra of BN-CNO/HA-DMPE at 2:1, 5:1, and 10:1 ratios in deionised water, where (**a**–**c**) are spectra at various concentrations, and (**d**–**f**) are spectra at various timepoints with a 100 µg/mL starting concentration.

**Figure 5 materials-15-05987-f005:**
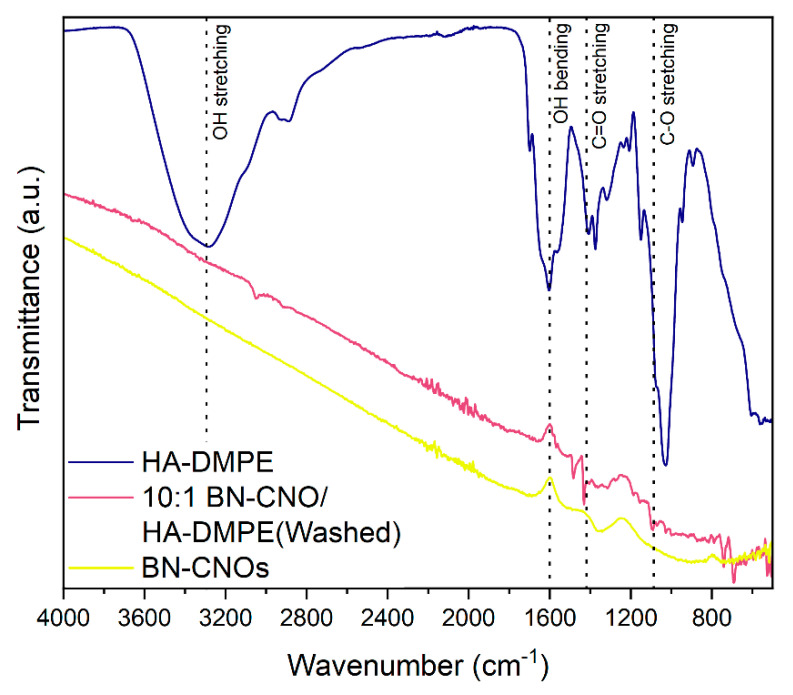
FTIR spectra of HA-DMPE, 10:1 BN-CNO/HA-DMPE (Washed), and BN-CNOs.

**Figure 6 materials-15-05987-f006:**
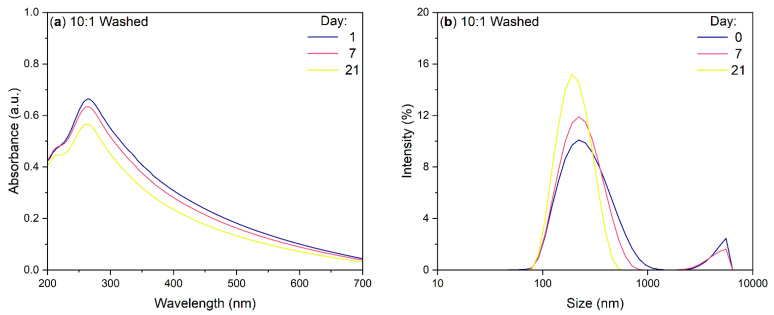
UV-Vis absorbance (**a**) and DLS (**b**) spectra of 10:1 BN-CNO/HA-DMPE washed sample at various times (solvent = deionised water; 100 µg/mL starting concentration).

**Table 1 materials-15-05987-t001:** Concentrations of 2:1, 5:1, and 10:1 BN-CNO/HA-DMPE water dispersions at various times.

Day	2:1 (µg/mL)	5:1 (µg/mL)	10:1 (µg/mL)	2:1 (µg/mL)	5:1 (µg/mL)	10:1 (µg/mL)
0	100	100	100	50	50	50
1	99	97	99	49	46	48
7	81	73	79	42	43	43
21	74	60	70	40	36	38
98	65	54	67	37	35	36
130	61	47	66	34	35	33
155	55	42	58	31	31	29
183	47	38	49	28	26	26

**Table 2 materials-15-05987-t002:** Mean hydrodynamic diameters and zeta potential (ZP) of 2:1, 5:1, and 10:1 BN-CNO/HA-DMPE at different concentrations.

(µg/mL)	2:1 (nm)	5:1 (nm)	10:1 (nm)	2:1 ZP (mV)	5:1 ZP (mV)	10:1 ZP (mV)
100	270	286	268	−28.2 ± 4.6	−32.4 ± 4.7	−30.1 ± 4.8
50	260	296	250	-	-	-
20	256	258	250	-	-	-
10	248	241	231	-	-	-
5	237	226	234	-	-	-

**Table 3 materials-15-05987-t003:** Mean hydrodynamic diameters of 2:1, 5:1, and 10:1 BN-CNO/HA-DMPE dispersions.

Day	2:1 (nm)	5:1 (nm)	10:1 (nm)
0	270	286	268
7	210	227	214
14	204	209	203
21	225	229	206

## Data Availability

The raw data from this study is available on request from the corresponding author.

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
