# Peer review of "Supramolecular Functionalisation of B/N Co-Doped Carbon Nano-Onions for Novel Nanocarrier Systems"

_materials, 2022, doi:10.3390/ma15175987_

Round 1

Reviewer 1 Report

This is a nice paper where authors report the non-covalent functionalization of B/N co-doped CNOs with the aim to increase their water dispersibility and stability for biological applications. The characterization of functionalized BN-CNOs by FT-IR, UV-vis, DLS and zeta potential seems indicate that the HA-DPE is interacting with the BN-CNO surface by a polymer wrapping mechanism. Authors write a nice introduction, the experimental part is well detailed and the results are cleaky written. The paper is worthy to be published in Materials.  

I have found two minor mistakes:

1)      In page 2, Figure 2 should be Figure 1.

2)      Please check the format of Ref. 4

Author Response

Dear Reviewer,

We thank you for positive comments.

In response to your comments and recommendations, we have revised the manuscript accordingly and we are resubmitting it for your further consideration.

Enclosed is a list of point-by-point responses to your comments. All the changes made in the manuscript are highlighted in yellow.

We believe that in its present form, the manuscript is suitable for publication in MDPI Materials.

Answer to comment 1: The referencing mistake on page 2 has been fixed; Figure 2 is now Figure 1.

Answer to comment 2: The format of Reference 4 was fixed.

Reviewer 2 Report

 1. In this study authors should clearly mention in the last paragraph of the Introduction why their nanocarrier is more advantageous than prior established nanocarriers. 

2. The B/N co-doped carbon nano-onions synthesized according to Ref "11" functionalized according to Ref. "23". So authors should be mentioned, which part of this work is new.

3. Figure-3, representation is not clear, why are authors telling these "celebrations"? In the manuscript, the authors should discuss more details of figure 3 results. like: What do they want to establish with this figure? why is it calibration? why it's needed etc? 

4. In figure 5, "Large aggregates (approx. 6,000 nm in diameter" Authors should provide evidence. Why the hydrodynamic sizes increase with increasing days, authors should more details discuss in the manuscript. Also, Figure 5 is not clear. 

5. Peak and stretch vibration should be denoted in Figure 6, it will give more impact on the readers as well as the manuscript. 

6. Authors should reconstruct all the figures distinctly. Mentioned their proper x and y axis and Figures should be properly labeled. 

7. In discussion and conclusion part should be more elaborate.

Author Response

Dear Reviewer, 

We thank you for the stimulating and insightful comments, which we believe have served to strengthen the quality of the manuscript.

In response to your comments and recommendations, we have revised the manuscript accordingly and we are resubmitting it for your further consideration. 

Enclosed is a list of point-by-point responses to your comments. All the changes made in the manuscript are highlighted in yellow. 

We believe that in its present form, the manuscript is suitable for publication in MDPI Materials.

Answer to comment 1: The paragraph at the end of the introduction has been expanded to include the advantages of our nanocarrier system. 

Answer to comment 2: Whilst some aspects of this work have been completed before, this manuscript represents the first non-covalent functionalisation of boron/nitrogen co-doped carbon nano-onions (BN-CNOs), which is novel. This work also includes long term stability studies (> 1 month) and FTIR analysis, which have not been done on a HA-DMPE/nano-onion system before. 

Answer to comment 3: Figure 2 has been reconstructed to make it clearer and its description in the results section has been expanded to explain it better. The purpose of this figure is to illustrate the long-term aqueous dispersibility of the nanoconjugate system, this is essential to know for future drug delivery applications. 

Answer to comment 4: Figure 4 has also been reconstructed to make it clearer. Evidence of the large aggregates can be seen in the DLS spectra (the small peak to the right of the main peak), the manuscript has been updated to clarify this. The hydrodynamic diameters of the particles decreased with increased stand time. 

Answer to comment 5: Thank you for your suggestion, we have labelled the FTIR peaks in figure 5. 

Answer to comment 6: As per your suggestion, figures 2 and 4 have been reconstructed to separate the individual spectra and label their individual x and y axis. 

Answer to comment 7: The conclusion and discussion parts of the manuscript have been improved and more material has been added. 

Reviewer 3 Report

In this work, boron/nitrogen co-doped carbon nano-onions were non-covalently functionalised with a hyaluronic acid-1,2-dimyristoyl-sn-glycero-3-phosphoethanola-mine conjugate to impart aqueous dispersibility and targeting affinity for the CD44 receptor. The nanocomposite exists as a static dispersion in water, as the conjugate polymer is permanently affixed to the surface of the doped nano-onions. It is recommended for publication after addressing the following questions.

Figure 1 and figure 2 can be integrated together as one image.

A typical TEM image of nanoparticles need be supplied in the manuscript.

Why the FTIR spectra of the three sample are so different?

Is there any potential application for this material? Some demonstration is encouraged to be added in the manuscript.

The references should be formatted and more 2022 year published works can be cited.

Author Response

Dear Reviewer, 

We thank you for the stimulating and insightful comments, which we believe have served to strengthen the quality of the manuscript.

In response to your comments and recommendations, we have revised the manuscript accordingly and we are resubmitting it for your further consideration. 

Enclosed is a list of point-by-point responses to your comments. All the changes made in the manuscript are highlighted in yellow. 

We believe that in its present form, the manuscript is suitable for publication in MDPI Materials.

Answer to comment 1: Your valuable feedback has been taken into account, figures 1 and 2 have been combined. 

Answer to comment 2: Unfortunately, we do not have a TEM instrument in Dublin City University. We are unable to send samples to collaborators to image as this would take weeks, causing long delays to the manuscript publication. 

Answer to comment 3: The first FTIR scan is of the HA-DMPE polymer alone and produces a “normal” looking spectrum. The second and third scans are of HA-DMPE functionalised BN- CNOs and pristine BN-CNOs, respectively. Due to the light-absorbing properties of the nano- onions, their resulting FTIR spectra have large slopes and small peaks. 

Answer to comment 4: We are currently working on drug delivery and cellular imaging applications for this nanocarrier. These experiments are still underway and the results will be published in a different manuscript. 

Answer to comment 5: The references have been formatted correctly and more works from 2022 have been cited. 

Round 2

Reviewer 2 Report

No comments